# Strength of carbon nanotubes depends on their chemical structures

Akira Takakura[1,2,7,8], Ko Beppu[3,8], Taishi Nishihara [1,2,7,8], Akihito Fukui[3], Takahiro Kozeki [4], Takahiro Namazu[3], Yuhei Miyauchi [1,2,5] & Kenichiro Itami [1,2,6]

Single-walled carbon nanotubes theoretically possess ultimate intrinsic tensile strengths in the 100–200 GPa range, among the highest in existing materials. However, all of the experimentally reported values are considerably lower and exhibit a considerable degree of scatter, with the lack of structural information inhibiting constraints on their associated mechanisms. Here, we report the first experimental measurements of the ultimate tensile strengths of individual structure-defined, single-walled carbon nanotubes. The strength depends on the chiral structure of the nanotube, with small-diameter, near-armchair nanotubes exhibiting the highest tensile strengths. This observed structural dependence is comprehensively understood via the intrinsic structure-dependent inter-atomic stress, with its concentration at structural defects inevitably existing in real nanotubes. These findings highlight the target nanotube structures that should be synthesized when attempting to fabricate the strongest materials.

[1] JST-ERATO, Itami Molecular Nanocarbon Project, Nagoya University, Chikusa, Nagoya 464-8602, Japan. [2] Graduate School of Science, Nagoya University, Chikusa, Nagoya 464-8602, Japan. [3] Department of Mechanical Engineering, Aichi Institute of Technology, Yakusa, Toyota, Aichi 470-0392, Japan. [4] Graduate School of Mechanical Engineering, University of Hyogo, Himeji, Hyogo 671-2201, Japan. [5] Institute of Advanced Energy, Kyoto University, Uji, Kyoto 611-0011, Japan. [6] Institute of Transformative Bio-Molecules (WPI-ITbM), Nagoya University, ChikusaNagoya 464-8602, Japan. [7] Present address: Institute of Advanced Energy, Kyoto University, Uji, Kyoto 611-0011, Japan. [8] These authors contributed equally: Akira Takakura, Ko Beppu, Taishi Nishihara. Correspondence and requests for materials should be addressed to T.Na (email: tnamazu@aitech.ac.jp) or to Y.M. (email: miyauchi@iae.kyoto-u.ac.jp) or to K.I. (email: itami@chem.nagoya-u.ac.jp)

High-strength and lightweight materials have always been highly sought after structural materials in a broad range of research fields, such as the fabrication of the safest and most fuel-efficient aircraft, or the construction of massive architectural structures. Single-walled carbon nanotubes (inset of Fig. 1a)[1], which can be viewed as cylindrically rolled graphene sheets[2,3], have been predicted as game-changing structural materials due to their outstanding theoretical strength per weight (strength-to-weight ratio; Fig. 1a)[4–13]. Ultimate intrinsic tensile strengths of more than 100 GPa[4–13] have been predicted. This extremely high-strength value, in combination with the lightweight nanotube structure, has even encouraged the construction of a space elevator (requires 63 GPa[14]), which is impossible using other existing materials. However, previous experimental studies have shown that the strength-to-weight ratio of real carbon nanotubes is typically a few times lower than the ideal case for defect-free single-walled carbon nanotubes[15–19], which is due to the existence of structural defects inevitably existing in real carbon nanotubes[7–13] and/or the inner walls of multi-walled carbon nanotubes that do not support the load directly[17,20]. Furthermore, the considerable degree of scatter among the measured samples[15–19] poses a critical problem regarding their practical use in macroscopic structural materials, such as yarns composed of many carbon nanotubes[21]. A recent study reported a considerable decrease in the net tensile strength of a carbon nanotube bundle as the number of included carbon nanotubes increased, presumably due to the heterogeneity of the individual carbon nanotubes, whose various structures yield non-uniform strengths, as well as the initial strain in each bundle[20]. The reason for the commonly observed nanotube-to-nanotube tensile strength variability is unclear, although their chiral structures are predicted to have considerable impacts on this observed variability[5–13], with the structures defined by either the chiral angle $\theta$ and diameter $d$, or the chiral indices $(n,m)$ (Fig. 1b)[22]. Despite tremendous pioneering efforts[15–20], there is still no experimental report on the correlation between the strength and chiral structure of single-walled carbon nanotubes due to the inherent difficulties in performing tensile tests on individual structure-defined, single-walled carbon nanotubes. This lack of a systematic experimental

study has long obscured the fracture mechanism of real single-walled carbon nanotubes, and therefore, has hindered the development of a macroscopic structural material with an ideal strength-to-weight ratio.

Here, we report the first direct measurements of the ultimate tensile strengths of individual structure-defined, single-walled carbon nanotubes (hereafter, referred to as nanotubes), providing clear insights into the strength and fracture toughness of nanotube structures. The strengths of the 16 measured nanotubes are in the 25–66 GPa range and are dependent on the chiral structure, with small-diameter, near-armchair nanotubes exhibiting the highest tensile strengths. This observed structural dependence is understood comprehensively via the intrinsic structure-dependent inter-atomic stress, together with its concentration at structural defects, which are virtually unavoidable in real nanotubes. The direction of the chemical bonds affects the net strength of the nanotubes primarily, and the concept of stress concentration, relying on classical linear elastic fracture mechanics, is still partially applicable. We successfully develop an empirical formula to predict the strengths of the real nanotubes, including unintentional structural defects. These findings clearly highlight the target nanotube structures to be synthesized, which are not well-constrained but may potentially be selectively grown, when attempting to fabricate the strongest macroscopic materials using carbon nanotubes.

## Results

**Tensile strength measurements of structure-defined nanotubes.** Figure 2 summarizes our experimental procedures. Individual nanotubes were synthesized over a micrometer-scale open slit via ambient alcohol chemical vapor deposition methods[23] that employed a modified fast-heating process (Fig. 2a)[24,25]. Broadband Rayleigh scattering spectroscopy was employed to determine the nanotube structures (Fig. 2b; see Methods and Supplementary Table 1)[26,27]. Then, the individual structure-defined nanotubes were picked up with a micro fork (Fig. 2c), and transferred onto a homemade microelectromechanical system (MEMS; Supplementary Fig. 1) device that was designed for the

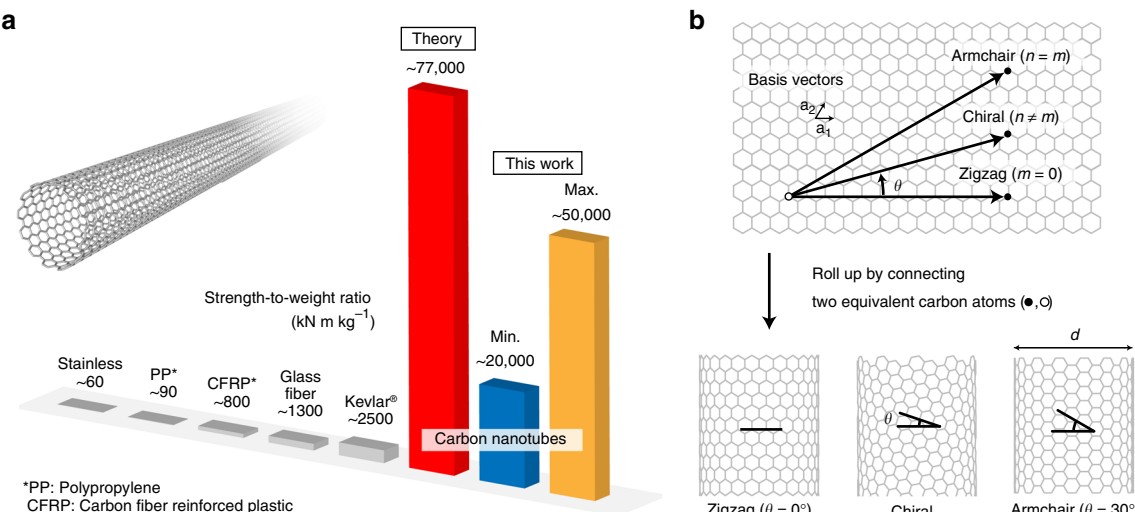

**Fig. 1** Single-walled carbon nanotubes. **a** Theoretical (red) and experimental (blue and yellow, representing the minimum and maximum values obtained in this study, respectively) strength-to-weight ratios of single-walled carbon nanotubes, compared with those of typical structural materials. The inset shows a single-walled carbon nanotube. **b** Classification of the single-walled carbon nanotubes by their chiral indices $(n,m)$, or diameter $(d)$ and chiral angle $(\theta)$. The chiral indices $(n,m)$ define the chiral vector (black arrows) that connects two equivalent carbon atoms in a graphene plane, and is represented as $n\mathbf{a}_1 + m\mathbf{a}_2$, where $\mathbf{a}_1$ and $\mathbf{a}_2$ are the basis vectors. The chiral angle is defined as the angle between the zigzag direction and circumference ($\theta = 0°$–$30°$). Achiral nanotubes, where $\theta = 0$ and $30°$, are called zigzag and armchair nanotubes, respectively

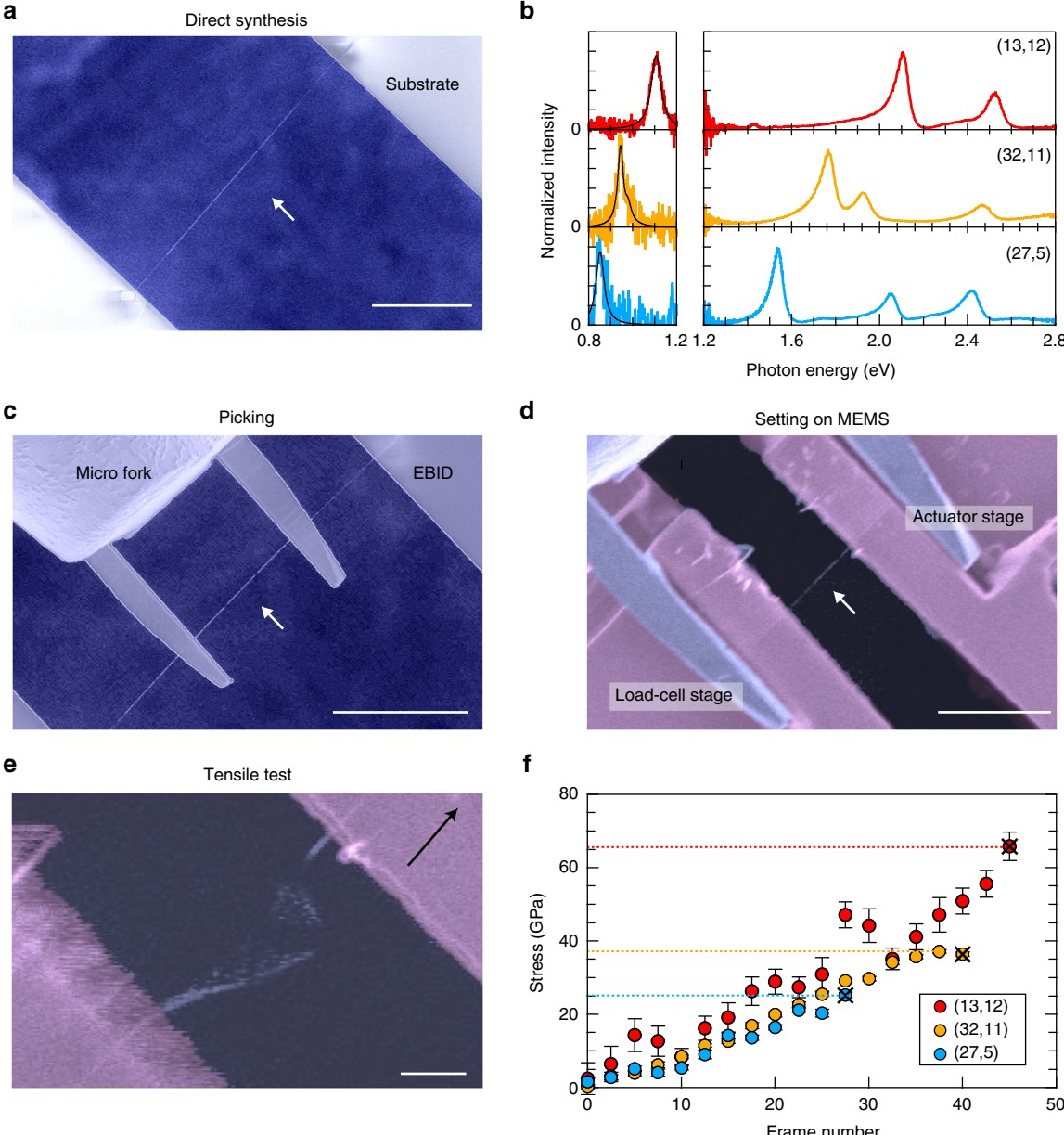

**Fig. 2** Experimental procedures for the tensile strength measurement. **a** Individual nanotube directly synthesized over an open slit. Scale bar, 10 μm. **b** Broadband Rayleigh scattering spectra of three nanotube species used for the chiral structure assignment. Different photodetectors were used for the 0.8–1.2 eV and 1.2–2.8 eV ranges. The black curves show the fitting results (see Methods). **c** Pick-up operation of an individual nanotube using a micro fork. Scale bar, 10 μm. **d** Setting a nanotube on the microelectromechanical system (MEMS) device. Scale bar, 5 μm. The nanotube is fixed to either the micro fork, substrate, or MEMS device via the electron-beam-induced deposition (EBID) method. **e** Image taken at the moment of nanotube fracture during the tensile test. The black arrow indicates the direction of actuator stage movement. Scale bar, 1 μm. **f** Stress as a function of image frame number for three nanotube species. The dashed lines indicate the tensile strengths. The color, contrast, and brightness of each image are tailored for clarity. The error bars indicate the 95% confidence levels

uniaxial tensile testing of small objects in a scanning electron microscope (SEM; Fig. 2d)[28]. Each individual nanotube was suspended and cramped between a pair of sample stages that were connected to a calibrated micro load-cell (left) and actuator (right) for the direct force measurement and uniaxial tensile force application, respectively (Fig. 2d). Figure 2e shows an image at the moment the nanotube fractured during tensile loading, with the central location of the nanotube fracture demonstrating that the nanotube was tightly fixed to the stages. The structure-dependent strengths were obtained using observable position markers on the edges of the stages to minimize any additional damage to the nanotube (the nanotube was out of the SEM field

of view during the measurements). The force was directly evaluated from the measured displacement of the load-cell stage equipped with U-shaped suspension beams (according to Hooke's law; see Methods for the details of the MEMS device). Figure 2f shows examples of the stresses applied to three structure-defined nanotubes as a function of image frame number to detect the load-cell stage positions (one frame per second; see Methods for determining the confidence levels). The nominal stress was evaluated using the cross-sectional area of the nanotube, $\pi dt$, where $t$ is the shell thickness, which is taken as the inter-layer graphite separation (0.34 nm)[15]. The highest stresses that were detected before fracture (indicated by the dashed lines)

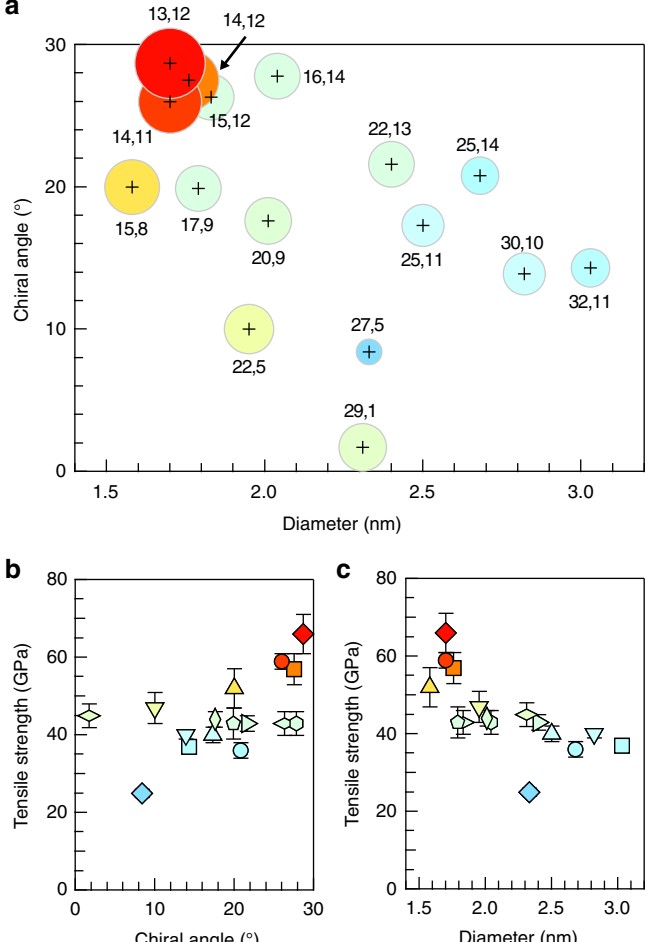

**Fig. 3** Chirality dependence of the ultimate tensile strength. **a** The tensile strengths of the chiral (n,m) structures are indicated by the circle diameters. The coordinates of the cross marks correspond to the chiral angle and diameter of each (n,m) nanotube. **b**, **c** The tensile strengths are plotted as a function of the chiral angle (**b**) and diameter (**c**). The color-symbol combinations of the points correspond to the measured nanotubes. The error bars indicate the 95% confidence levels

correspond to the ultimate tensile strengths. If we assume that the nanotube does not slip on the stages, the fracture strain of a nanotube with a strength of ~50 GPa (Supplementary Fig. 2) is estimated as ~5%, suggesting a Young's modulus of ~1 TPa, which is consistent with reported values[29].

**Structure dependence of the ultimate tensile strength**. We succeeded in measuring the tensile strengths of 16 structure-defined nanotube species in this study. Figure 3a summarizes the chirality dependence of the measured ultimate tensile nanotube strengths (see also Supplementary Table 2). The strengths are seemingly dependent on both the chiral angle (Fig. 3b) and diameter (Fig. 3c) of the nanotubes. The tensile strengths are in 25–66 GPa range, which is several times smaller than the theoretical predictions for ideal pristine nanotubes[4–13]. This suggests that the observed fractures are dominated by extrinsic factors, most likely consisting of structural defects, such as atomic vacancies[7–10], topological defects[12], or helical structural defects[13] on the nanotubes. Stone–Wales[11] defects may be ruled out as the responsible defect for the nanotube fracturing observed in this study because the strength reduction due to the Stone–Wales defect was only predicted to be 20–30%. Although we synthesized

high-quality nanotubes that exhibited negligible defect-derived D-mode signals in their Raman spectra (Supplementary Fig. 3), it is still highly probable that the nanotubes had small numbers of structural defects within the >5-μm-long test segments, which include ~$10^6$ carbon atoms.

## Discussion

Let us now discuss the implications of the results on the nanotube fracture mechanisms. We make three fundamental assumptions for the analysis as follows: (i) the C–C bond breaks when the stress applied to the bond exceeds a certain value, regardless of the nanotube structure; (ii) the stress concentration occurs at the defect crack edges[10,13], and (iii) brittle fracture occurs once a C–C bond in the weakest defect crack breaks. We first consider the factor that may dominate the chiral angle dependence. When a net uniaxial stress, $\sigma$, is applied to a nanotube (Fig. 4a), the effective stress applied along the C–C bonds that are approximately along the nanotube axis, $\sigma_{CC}$, is the highest (the blue-colored bonds and all of the equivalent ones in Fig. 4a), and should depend on the chiral angle, $\theta$, of the nanotube due to the difference in the nanotube axis and C–C bond directions[6]. This effective inter-atomic stress, $\sigma_{CC}$, is approximately related to $\sigma$ as $\sigma_{CC} = f(\theta)\sigma$, where $f(\theta) = (1/2)[(1-\nu)+(1+\nu)\cos 2\theta]$ (see Supplementary Note 1), assuming homogeneous deformation ($\nu = 0.16$ is the Poisson ratio of graphite)[6] and a Young's modulus that is independent of the chiral structure[30]. Therefore, the C–C bonds effectively feel different stresses by a factor $f(\theta)$, even when the same $\sigma$ is applied to the nanotubes. We then further consider the stress concentration at the defect crack edges, as has been predicted in recent theoretical studies[10,13] on nanotubes with various defects (Fig. 4a). Since the stress concentration should depend on the size and shape of the defects, which are unknown, we define the stress concentration factor, $K(d)$, as an empirical parameter that scales according to the tube diameter, where $K(d) \propto d^\alpha$. This allows the inter-atomic stress at a defect crack edge, $\sigma_{CC}^*$ (on the red-colored bonds in Fig. 4a), and $\sigma$ to be related as $\sigma_{CC}^* = K(d)f(\theta)\sigma$. Finally, since assumption (i) states that the maximum $\sigma_{CC}^*$ at the moment of bond fracture is constant, we obtain an empirical formula that relates the ultimate tensile strength, $\sigma_f$, to $d$ and $\theta$ as:

$$\sigma_f = Cf(\theta)^{-1}d^{-\alpha}, \tag{1}$$

where $C$ and $\alpha$ are empirical factors that depend on the details of the potential nanotube defects.

We determined $\alpha$ by plotting the product of the tensile strength and $f(\theta)$ as a function of diameter (Supplementary Fig. 4). This approach allows us to extract only the diameter dependence of the tensile strength with excluding the impact of the chiral angle dependence on the effective stress along the C–C bonds by a factor of $f(\theta)$. The diameter dependence of $\sigma_f f(\theta)$ is found to be well described by $\sigma_f f(\theta) \propto d^{-\alpha}$, where $\alpha = 0.5 \pm 0.2$ yields the best fit to the data. This result suggests that $f(\theta)d^{0.5}$ adequately describes the obtained data, as shown in Fig. 4b. Finally, we find that Eq. (1) reproduces all of the experimental results when $\alpha = 0.5$ and $C = 55 \pm 2$ GPa nm$^{0.5}$. This successful fit clearly suggests that the inter-atomic stress between the C–C bonds inherently depends on the chiral angle[6]. Furthermore, the $1/\sqrt{d}$ dependence determined by our best fit also highlights an important implication. A recent theoretical study reported that the fracture strength of nanotubes with various types of defects should exhibit a universal dependence on the defect length, $a$, along the circumference direction as $\sigma_f \propto 1/\sqrt{\pi a}$, regardless of the detailed defect structures, which conforms to classical linear elastic fracture mechanics[10]. Here, the $1/\sqrt{d}$ dependence indicates a linear relationship between $d$ and $a$ of the weakest defect crack

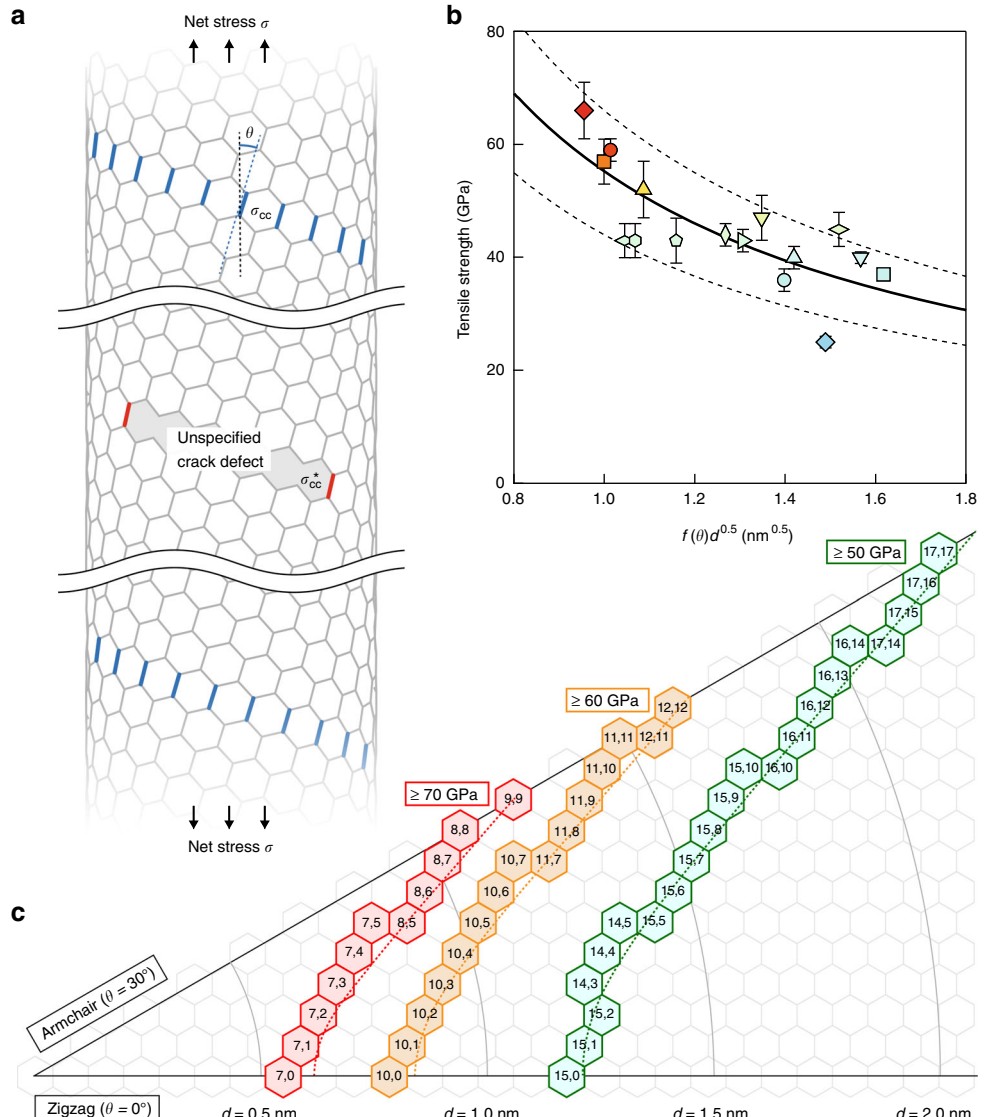

**Fig. 4** Empirical modeling of nanotube tensile strength. **a** Net uniaxial stress, $\sigma$, effective stress applied on the C-C bonds, $\sigma_{CC}$, at angle $\theta$ against the nanotube axis direction (blue), and concentrated stress, $\sigma_{CC}^*$, on the C-C bonds (red) at the defect crack edges. **b** Tensile strength plotted as a function of $f(\theta)\sqrt{d}$, where $\theta$ and $d$ are the chiral angle and diameter, respectively. $f(\theta)$ is given by $(1/2)[(1-v)+(1+v)\cos 2\theta]$ ($v = 0.16$ is the Poisson ratio of graphite). The solid curve is the best fit to the data, which is $C[f(\theta)\sqrt{d}]^{-1}$, where $C = 55$ GPa nm$^{0.5}$. The dotted curves indicate the ±20% range ($C = 44$ and $66$ GPa nm$^{0.5}$ for the lower and upper curves, respectively). The error bars indicate the 95% confidence levels. **c** Empirical contour map of the tensile strengths. The red, yellow, and green regions show the chiral $(n,m)$ structures, with predicted strengths of approximately 70, 60, and 50 GPa, respectively

responsible for nanotube fracture. Although the origin of this dependence is still unclear, it may be reasonable that the size of the maximum defect (where the nanotube is weakest) is limited by the circumference length, $\pi d$. Therefore, we conclude that the real nanotube strength is determined by both the intrinsic chiral angle dependence, $f(\theta)$, and the extrinsic stress concentration at the edge of the largest (weakest) defect crack, whose size along the circumference is proportional to the diameter.

Finally, we comment briefly on the practical impact of the findings in this study. Figure 4c shows the predicted strengths of the nanotubes via Eq. (1) (see also Supplementary Table 3 for the list of the estimated structure-dependent fracture strengths of the $(n,m)$ nanotubes via Eq. (1)). Various types of nanotubes may exhibit tensile strengths above ~60 GPa, which is known as the minimum threshold requirement for constructing a space elevator[14]. Since the requirement for the structure selectivity to achieve strengths > 60 GPa is not well-constrained, it seems feasible to fabricate long, well-organized bundles[20] consisting of

select structure-grown nanotubes[31] within the target range of $(d,\theta)$ in the near future. Therefore, our findings highlight the target chemical structures that should be selectively synthesized for the realization of these high-strength structural materials, and also provide a comprehensive understanding of the fracture mechanism of real carbon nanotubes, which may potentially lead to the development of new methods to overcome current real carbon nanotube strength limits.

## Methods

**Structure assignment via broadband Rayleigh spectroscopy.** Each nanotube has an inherent series of multiple optical transitions that originates from the exciton resonances in each one-dimensional subband[26,32–34] (Fig. 2b), whose correspondence to potential nanotube structures $(d,\theta)$ or $(n,m)$ has been well-established in empirical tables[35]. The energies of the optical transitions are widely distributed across the infrared-to-visible photon energy range. Rayleigh spectroscopy has been used previously as a powerful method to probe the optical transitions in the photon energy range above 1.2 eV, which is limited by the detectable range of commonly used silicon-based detectors[33,35–40]. Since the lack of spectral

information below 1.2 eV often yields uncertainties in the nanotube structure assignment[35,38,39], we expanded the detection range to 0.8 eV using a near-infrared detector, which reduced the uncertainties and yielded considerable improvements in the accuracy and efficiency of the structure assignment process (broadband Rayleigh spectroscopy)[27]. Supplementary Fig. 5 shows a schematic of the broadband Rayleigh spectroscopy approach. The broadband light from a supercontinuum source (Fianium, WL-SC-400-PP-4 or YSL photonics, SC-Pro) was focused on an individual nanotube that was placed in a vacuum chamber. The integrated power was ~ 2 mW for photon energies in the 0.56–2.8 eV range. The scattered light collected through an objective lens (numerical aperture of 0.42) was detected using one of the following: a charge-coupled device (CCD) camera for imaging, a monochromator with a thermoelectrically cooled silicon CCD camera (Princeton Instruments, ProEM; 1.2–2.8 eV), or a monochromator with a thermoelectrically cooled indium-gallium-arsenide camera (Princeton Instruments, NIRvana; 0.8–1.4 eV). The Rayleigh scattering cross-section is proportional to $\omega^3|\chi(\omega)|^2$, where $\omega$ and $\chi(\omega)$ are the optical frequency and susceptibility, respectively[36]. Each Rayleigh spectrum was corrected for the $\omega^3$ scattering efficiency factor to show the optical susceptibility. We analyzed the excitonic response of each nanotube using a Lorentzian line shape of the form $\chi(\omega) = \chi_b + f[(\omega_0-\omega)-i\Gamma/2]^{-1}$, where $\chi_b$, $f$, $\omega_0$, and $\Gamma$ are the (frequency-independent) background susceptibilities arising from nonresonant transitions, the exciton oscillator strength, the resonant frequency, and the linewidth, respectively.

**Micro fork and microelectromechanical system (MEMS) device.** Silicon or tungsten micro forks were fabricated using the focused ion beam method. Supplementary Fig. 1a shows a SEM image of the MEMS tensile test device designed for small objects[28]. This device consists of sample stages, a comb-drive electrostatic actuator for generating the uniaxial tensile force, and capacitive sensors (not used in this work), with electric insulation between the sensors to avoid unexpected electric effects (Supplementary Fig. 1b). The comb structure and sample stages are supported by U-shaped suspension beams, which can freely move in the in-plane tensile direction. The sample stage connected to the actuator is referred to as the actuator stage, and the other stage is referred to as the load-cell stage in the main text. The spring constant is determined on the four U-shaped suspension beams (indicated by the orange color in Supplementary Fig. 1b). We moved the load-cell stage using a calibrated microforce sensing probe (FEMTO TOOLS, FT-S1000-LAT) to calibrate the spring constant, and measured the force and displacement of the load-cell stage.

**Determination of the error bars**. All of the stress measurements were recorded by the SEM observations of the stage positions, which were evaluated by the SEM image analysis. The SEM images were affected by the electronic noise arising from the stage actuation, which introduced a degree of scatter to the detected marker positions. Therefore, we conducted a statistical analysis of the detected positions in a 1-s timeframe, and plotted the stresses at each timeframe, which were calculated using the average stage positions (solid symbols), with the 95% confidence level error bars shown in Figs. 2f, 3b, c, 4b, and Supplementary Fig. 4.

## Data availability
The data that support the findings of this study are available from the corresponding author upon reasonable request.

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

## Acknowledgements

This work was supported by the ERATO program from JST (K.I.) (grant number: JPMJER1302). Part of this work was supported by JSPS KAKENHI grant number JP24681031 (Y.M.). We thank Keisuke Matsui (Nagoya University) for assistance in synthesizing and locating the nanotubes.

## Author contributions

Y.M. conceived the concept and K.I. directed the project. A.T. fabricated the nanotube synthesis apparatus and synthesized the nanotubes. T.Ni. and A.T. arranged and carried out the optical measurements. T.Na., T.K., and K.B. developed the MEMS devices and handling methods of the individual nanotubes. A.T., K.B., and A.F. performed the tensile tests. Y.M. and T.Ni. considered the fracture mechanism. All the authors contributed to writing the paper.

## Additional information

**Competing interests:** The authors declare no competing interests.

