## [Peer Review File · Nature Communications]

Reviewers' comments:

Reviewer #1 (Remarks to the Author):

In this manuscript, A. Takakura et al. described the measurement of the tensile strengths for structure-defined individual carbon nanotubes. The experimental setup consists of a homemade microelectromechanical system (MEMS) device, which was previously used to measure the uniaxial tensile testing of Si nanowires [refs 28], and now allows for the ultimate tensile strength of carbon nanotubes in a scanning electron microscope. Based on the experimental data of sixteen chirality-determined SWNT, they further developed an empirical formula to predict the strength of other kinds of SWNTs. These results will be useful for real nanotube applications and can be helpful for understanding the inconsistent CNT tensile strength of previous studies. I do find many issues in the manuscript need to be further considered carefully before it can be considered for publication. These specific comments are listed as below.

1. In Figure 2c, the word "Micro folk "should be "Micro fork ". This typo also exists in the corresponding text.
2. In Page 6, the specific derivation of $f(\theta)$ should be given in the supplementary material.
3. In the text, experimental strength-to-weight ratio of SWNT was compared with those of typical structural materials. How is the measured ultimate atomic strength of CNT compared to other graphitic materials, such as graphene oxide fibres?
4. Is there any other evidence that the fracture occurred at the defect? Is it possible to provide a Raman D-mode mapping along the carbon nanotube? If purposely changing the concentration of defects, for example, irradiation processes or thermal annealing, would affect the ultimate tensile strength? Only the type of defect matters?
5. Regarding references, some important references (i.e. seminal works...) are missing. Especially the early optical papers should be included. Here are some (among others): (Science, 2002, 298, 2361); (Science, 2004, 306, 1540); (Nano letter, 2011, 11, 1); (Nature Nanotechnology, 2013, 8, 917); (PNAS, 2014, 111, 7564) and (Nano Research, 2015, 8, 303) should at least be cited.

Reviewer #2 (Remarks to the Author):

The data and conclusion are interesting and topical, but not revolutionary in the study of CNT. The presentation, however, has holes that need to be filled or repaired. The main issue is the lack of details on the statistical analysis of the experimental data and the fitted equation for the strength of the various CNT.

Supplement fig 3 (cited on p6) is not included in the materials and hence there is no clear sense of the accuracy of chirality assignments for the individual CNT.

There is no discussion of the uncertainty in measurements of the strain. (ie how are the error bars in figs 2,3, & 4 calculated?)

As the CNT are being observed visually as they are strained, why is the breaking strain only "estimated at 5%" for all CNT instead of measured for each individual sample.

Can the authors comment on the possibility of damage caused by the manipulation of the CNT from one holder to the other? Or damage from clamping in the MEMS device?

No details are given for the quoted uncertainty in the exponent in the strain in equation 1. How is the fit in fig 4b obtained? What method yielded the fit? It appears that the authors simply plot the data against $f(\theta)d^{-0.5}$ because they think/hope that that this is the correct model. If this was the method, then the assertion of any particular uncertainty in the exponent is difficult to justify.

Also eq 1 is not new and the authors should cite prior work (eg ref 10, etc).

The authors may also wish to contrast their conclusions with known results from macroscopic woven cables, etc, but this would be a matter of taste.

Surely the authors intended to say "micro-fork" (small tined implement) instead of "micro folk" (tiny groups of people)? There are a few other places where the rules of grammar and spelling are contravened.

Responses to the Reviewers' comments and suggestions

We are grateful for the reviewers' comments and suggestions for improving the manuscript. Using the reviewers' insightful comments, we have revised the manuscript. In the following, point-by-point responses to the reviewers' comments and the revisions included in the revised manuscript are presented.

For Reviewer #1:

Remarks to the Author:

In this manuscript, A.Takakura et al. described the measurement of the tensile strengths for structure-defined individual carbon nanotubes. The experimental setup consists of a homemade microelectromechanical system (MEMS) device, which was previously used to measure the uniaxial tensile testing of Si nanowires [refs 28], and now allows for the ultimate tensile strength of carbon nanotubes in a scanning electron microscope. Based on the experimental data of sixteen chirality-determined SWNT, they further developed an empirical formula to predict the strength of other kinds of SWNTs. These results will be useful for real nanotube applications and can be helpful for understanding the inconsistent CNT tensile strength of previous studies. I do find many issues in the manuscript need to be further considered carefully before it can be considered for publication. These specific comments are listed as below.

We thank the reviewer for evaluating our works as useful and helpful. Below are our responses.

Q1-1) *In Figure 2c, the word "Micro folk" should be "Micro fork". This typo also exists in the corresponding text.*

Thank you for finding our misspellings. We have revised the misspelled "Micro folk" in the manuscript.

Q1-2) *In Page 6, the specific derivation of $f(\theta)$ should be given in the supplementary material.*

$f(\theta)$ in Eq. (1) of the main text is derived from the coordinate transformation of the stress tensor. A stress in a two-dimensional plane is described as a second-order tensor of the form:

$$\varepsilon = \begin{pmatrix} \varepsilon_{11} & \varepsilon_{12} \\ \varepsilon_{21} & \varepsilon_{22} \end{pmatrix}.$$

If the coordination system is rotated by an angle θ (Fig. R1), the stress tensor in the new coordination system is related to the original one as:

$$\varepsilon' = \begin{pmatrix} \varepsilon_{11}' & \varepsilon_{12}' \\ \varepsilon_{21}' & \varepsilon_{22}' \end{pmatrix} = \begin{pmatrix} \cos \theta & \sin \theta \\ -\sin \theta & \cos \theta \end{pmatrix} \begin{pmatrix} \varepsilon_{11} & \varepsilon_{12} \\ \varepsilon_{21} & \varepsilon_{22} \end{pmatrix} \begin{pmatrix} \cos \theta & -\sin \theta \\ \sin \theta & \cos \theta \end{pmatrix}.$$

In our discussion, the x -axis in the original coordination system (x - y) and the x' -axis in the new one (x' - y') are parallel to the tube axis and C-C bond direction, respectively. Therefore, under the uniaxial strain condition along the nanotube axis ($\varepsilon_{12} = \varepsilon_{21} = 0$), ε_{11}' , the stress along the C-C bond direction, is given as

$$\begin{aligned}\varepsilon_{11}' &= \varepsilon_{11} \cos^2 \theta - \nu \varepsilon_{11} \sin^2 \theta \\ &= \frac{\varepsilon_{11}}{2} [(1 - \nu) + (1 + \nu) \cos 2\theta] \\ &= \varepsilon_{11} f(\theta),\end{aligned}$$

where ν is the Poisson ratio and $\varepsilon_{22} = -\nu \varepsilon_{11}$. We added a sentence at page 6, 11th line of 2nd paragraph, as “see Supplementary Note 1”, and the detailed description on the derivation of $f(\theta)$ in the supplementary material (Supplementary Note 1).

Figure R1. The basis vectors in the x - y and the x' - y' coordination systems are parallel to a tube axis and a C-C bond direction, respectively.

Q1-3) *In the text, experimental strength-to-weight ratio of SWNT was compared with those of typical structural materials. How is the measured ultimate atomic strength of CNT compared to other graphitic materials, such as graphene oxide fibres?*

Because carbon nanotubes are cylinders of graphene, their fundamental ultimate atomic strength should be on the similar order to that of graphene (130 GPa (Ref. 3)). With regard to graphene oxide fibers, although many efforts have been devoted for improving the mechanical strength of graphene oxide fibers, their strength is on the order of 100–500 MPa (RSC Adv., 2019, **9**, 4198; Sci. Rep. 2018, **8**, 10803), which is much weaker than individual singled-walled carbon nanotubes (25–66 GPa in this work).

Q1-4a) *Is there any other evidence that the fracture occurred at the defect?*

At this stage, only the fact that the measured strengths of all nanotubes were lower than the theoretical predictions (~ 100 GPa) is the indirect evidence of the defect-induced fracture.

Q1-4b) *Is it possible to provide a Raman D-mode mapping along the carbon nanotube?*

Although it should be possible technically, we have not performed the Raman *D*-mode mapping. This is mostly because we have never obtained Raman spectra with identifiable *D*-mode signal from more than 30 suspended nanotubes grown and ever measured in our group. Thus, we concluded that the defect density in our nanotubes is too small to be detected using a standard far-field Raman spectroscopy.

Q1-4c) *If purposely changing the concentration of defects, for example, irradiation processes or thermal annealing, would affect the ultimate tensile strength? Only the type of defect matters?*

Thank you for this comment. We have also considered that answering the above questions is important, and we have been trying to purposely generate sparse defects with defined types to test the effects of artificially generated defects. However, at this point, we did not succeed in generating defects in a well-controlled manner and evaluating the defect density quantitatively. We believe this topic will be one of the next important challenges.

Q1-5) *Regarding references, some important references (i.e. seminal works...) are missing. Especially the early optical papers should be included. Here are some (among others): (Science, 2002, 298, 2361); (Science, 2004, 306, 1540); (Nano letter, 2011, 11, 1); (Nature Nanotechnology, 2013, 8, 917); (PNAS, 2014, 111, 7564) and (Nano Research, 2015, 8, 303) should at least be cited.*

We thank the reviewer for finding the missing of some important references. We added the references suggested by the reviewer (No. 32, 33, 34, 37, 39, and 40).

Reviewer #2 (Remarks to the Author):

The data and conclusion are interesting and topical, but not revolutionary in the study of CNT. The presentation, however, has holes that need to be filled or repaired. The main issue is the lack of details on the statistical analysis of the experimental data and the fitted equation for the strength of the various CNT.

We thank the reviewer for finding our paper of interest. According to the Reviewer's insightful suggestions, we have improved the manuscript. Below are our responses.

Q2-1) *Supplement fig 3 (cited on p6) is not included in the materials and hence there is no clear sense of the accuracy of chirality assignments for the individual CNT.*

We showed Supplementary Figure 3 only for demonstrating negligible Raman *D*-mode signal of our typical nanotube samples, but not for chirality assignments. Instead of using Raman spectroscopy, we used near-infrared-extended broadband Rayleigh spectroscopy that enables precise chirality assignment as described in the Methods section. We added the table of the observed optical resonance energies used for the chirality assignment as **Supplementary Table 1** to provide a clear sense of the accuracy of chirality assignments for the individual nanotubes to the readers.

Q2-2) *There is no discussion of the uncertainty in measurements of the strain. (ie how are the error bars in figs 2,3, & 4 calculated?)*

Thank you for this important comment. We are sorry about the lack of detailed procedure of the determination of the force and the uncertainty of it. In this study, we directly evaluated the force from the displacement of the load-cell stage with U-shaped suspension beams (as a spring with a known spring constant) using the Hooke's law. Hence, unlike some of the previous studies that used the strain and Young's modulus (e.g. Ref. 19) to estimate the force, the value of the strain is not directly used for the calculations of the stress and the error bars in this study. We added a sentence **at page 5, line 9 in the first paragraph**, as **"The force was directly evaluated from the measured displacement of the load-cell stage equipped with U-shaped suspension beams (according to Hooke's law; see Methods for the details of the MEMS device)."** to make this point clear.

In the measurements of the stress, the SEM images of the position markers prepared on the edges of the load-cell and the actuator stages were recorded; the displacement of the markers were evaluated based on the

SEM image analyses. In order to minimize the effect of random electronic noise arising from the stage actuator on the evaluation of the displacement from the SEM images, we conducted the statistical analysis of the detected positions, and plotted the average values with the 95 % confidence levels as error bars in Figs. 2f, 3b, 3c, 4b, and Supplementary Fig. 4. We added a sentence at page 5, line 13, as “see Methods for determining the confidence levels”. We also added a subsection in the Methods at page 10, as “**Determination of the error bars.** All of the stress measurements were recorded by the SEM observations of the stage positions, which were evaluated by the SEM image analysis. The SEM images were affected by the electronic noise arising from the stage actuation, which introduced a degree of scatter to the detected marker positions. Therefore, we conducted a statistical analysis of the detected positions in a one-second timeframe, and plotted the stresses at each timeframe, which were calculated using the average stage positions (solid symbols), with the 95% confidence level error bars shown in Figs. 2f, 3b, 3c, 4b, and Supplementary Fig. 4.”. We also added a sentence at page 18 (1st line from the bottom), at page 19 (1st line from the bottom), at page 21 (1st line), and at page 6 (1st line from the bottom) in Supplementary Information as “The error bars indicate the 95% confidence levels.”.

Q2-3) *As the CNT are being observed visually as they are strained, why is the breaking strain only "estimated at 5%" for all CNT instead of measured for each individual sample.*

We are sorry that the description at Page 5 in the previous manuscript “The typical strain at fracture was estimated at 5% for nanotubes with a strength of ~50 GPa (Supplementary Fig. 2), which suggests a Young’s modulus of ~1 TPa, which is consistent with reported values.” was misleading.

Although we showed visually an example that the fracture occurs near the center of the test segment (e.g. Fig. 2e), we have never observed the nanotubes themselves using the SEM during the measurements of the tensile stresses to avoid any possible effects of the electron beam irradiation (defect generation or bond rearrangement, etc.), as we described in Page 5 (line 7) “The structure-dependent strengths were obtained using observable position markers on the edges of the stages to minimize any additional damage to the nanotube (the nanotube was out of the SEM field of view during the measurements).”.

Thus, we measured the displacements between the load cell and the actuator stages, but not the strain of the nanotubes directly. The resultant relation between the determined stress and the strain estimated from the change of the distance between the two stages yielded Young’s modulus of about 1 TPa (typical value for a nanotube reported in many previous studies) as shown in Supplementary Fig. 2, and this implies that the distance between the two stages could be correlated to the actual strain of the nanotubes. However, it is still

difficult to completely rule out the possibility of the slight slipping of the nanotubes on the stages during the measurements. Therefore, as we described above, in this study we mainly focused on the stresses applied on the nanotubes that can be directly obtained from the displacement of the spring (U-shaped suspension beams of the MEMS device) connected to the nanotube with higher accuracy than the strains, and showed Supplementary Fig. 2 as a typical example of the relation between the stress and the displacement between the two stages (corresponds to strain when the slipping is negligible).

In order to make this point clear, we changed the horizontal axis label of Supplementary Fig. 2 as “ $\Delta L/L$ ”, where L is the initial distance between the two stages and ΔL is the displacement, and added/modified sentences in the caption of Supplementary Fig. 2 as “ L is the initial distance between the two stages, and ΔL is the displacement, such that $\Delta L/L$ is equal to the nanotube strain when the nanotube does not slip on the stages. The red line is the linear fit to the data, yielding a Young’s modulus of 1.05 ± 0.08 TPa, assuming no slip at the nanotube-stage interfaces.”.

Accordingly, we also modified the sentence at the last of the first paragraph of page 5 as “If we assume that the nanotube does not slip on the stages, the fracture strain of a nanotube with a strength of ~ 50 GPa (Supplementary Fig. 2) is estimated as $\sim 5\%$, suggesting a Young’s modulus of ~ 1 TPa, which is consistent with reported values²⁹.”.

Q2-4) *Can the authors comment on the possibility of damage caused by the manipulation of the CNT from one holder to the other? Or damage from clamping in the MEMS device?*

Unfortunately, it was difficult to precisely evaluate the damage of the nanotubes in the process of their manipulation in the current experimental scheme. However, because of the observations of the fracture near the center of the nanotube (see Fig. 2e), we consider that the clamping on the MEMS device itself had no critical effect for weakening the strengths of the nanotubes.

Q2-5) *No details are given for the quoted uncertainty in the exponent in the strain in equation 1. How is the fit in fig 4b obtained? What method yielded the fit? It appears that the authors simply plot the data against $f(\theta)d^{0.5}$ because they think/hope that that this is the correct model. If this was the method, then the assertion of any particular uncertainty in the exponent is difficult to justify.*

We thank the reviewer for this important comment. As suggested by the reviewer, our previous discussion was unclear owing to the lack of the description of the detailed fitting methods in the main text, and we improved the manuscript according to the suggestion of the reviewer as follows.

As we described in the Methods section of the previous manuscript, to begin with, we consider the chiral angle dependence to understand the structure (diameter d and chiral angle θ) dependence of the tensile strength. As we added the detailed derivation of $f(\theta)$ according to the Reviewer 1's comment (see Q1-2), the difference between the directions of the tensile axis and the C-C bond are taken into account by the transformation of the coordination. The same expression was also proposed in the previous work (Ref. 6). Secondly, we examined the diameter dependence of the tensile strength after excluding the expected chiral angle dependence (which is expressed by $f(\theta)$) of the tensile strength (see Supplementary Fig. 4); we multiplied the tensile strength with $f(\theta)$. Then we fitted the tensile strength as the power law of diameter (d^α), and we obtained the power index $\alpha = 0.5 \pm 0.2$ empirically. This result suggests that $f(\theta)d^{0.5}$ can be a good variable for describing the obtained data, and then we plotted the tensile strength as a function of $f(\theta)d^{0.5}$ in Fig. 4b. Thus, the derivation of $f(\theta)d^{0.5}$ does not come from our hopes, but it is based on the above analyses. To make this point clear, we moved the description of the fitting procedure to the main text (Page 7, 2nd paragraph) and added some words in the same paragraph.

Q2-6) *Also eq 1 is not new and the authors should cite prior work (eg ref 10, etc).*

As we explained above, we empirically introduced the diameter dependence of the tensile strength as the power law (d^α) in the Eq. (1). If we set $\alpha = 0.5$ from the beginning, it is appropriate to refer to Ref. 10 etc. as the direct source of Eq. (1). However, in this study α in Eq. (1) is introduced as a free parameter in the general empirical equation that can express any power law diameter dependence in the experimental result, if any, and $\alpha = 0.5 \pm 0.2$ was obtained purely from the fitting to the experimental result regardless of any assumption (or our hope) on the diameter dependence (Supplementary Fig. 4). Because it matches with the previous predictions (e.g. Ref. 10), we eventually discussed the experimental results in the context of the classical linear elastic fracture mechanics previously discussed in Ref. 10, etc. Since equations proposed in Ref. 10 (or Ref. 13, which also proposed a square-root law based on the different mechanism) are constructed on the premise that the power index of the diameter is 0.5, we still think that it is more appropriate that these studies are cited as a motivation to introduce the empirical power law of the diameter as we described in the main text (2nd paragraph of Page 6).

Q2-7) *The authors may also wish to contrast their conclusions with known results from macroscopic woven cables, etc, but this would be a matter of taste.*

The main scope of this study is rather to clarify structure (chiral angle and diameter) dependence of the strength of individual nanotubes that has been theoretically predicted but never confirmed experimentally. If one can make an ideal macroscopic woven cable in which all the individual components are near-armchair nanotubes and very long, in principle, the cable composed of such nanotubes can be stronger than the currently available ones according to the results in this study. However, the strength of the macroscopic cable is not only determined by that of individual components, but also by many other factors (length of each nanotube, jointing strength between individual nanotubes, etc). Thus, in this manuscript aiming at providing fundamental knowledge of the structure dependence of individual components, we refrain from contrasting our conclusions with known results from macroscopic woven cables to avoid any misleading.

Q2-8) *Surely the authors intended to say "micro-fork" (small tined implement) instead of "micro folk" (tiny groups of people)? There are a few other places where the rules of grammar and spelling are contravened.*

Thank you for finding our misspellings. Our manuscript has got English proofreading, and we revised the misspellings and grammars.

Once again, we would like to express our great appreciation to all the reviewers for providing many constructive comments, contributing to improve our manuscript significantly. We hope the revised manuscript will be accepted for publication and I sincerely look forward to hearing from you in the near future.

REVIEWERS' COMMENTS:

Reviewer #1 (Remarks to the Author):

The authors have carefully and sufficiently addressed the points raised by reviewers. The data and information added to the manuscript during revision made it significantly more clearer. I recommend the acceptance of the current manuscript version.

Reviewer #2 (Remarks to the Author):

The authors have replied in good faith to the criticisms from the referees.

Responses to the Reviewers' comments

For Reviewer #1 (Remarks to the Author):

The authors have carefully and sufficiently addressed the points raised by reviewers. The data and information added to the manuscript during revision made it significantly clearer. I recommend the acceptance of the current manuscript version.

For Reviewer #2 (Remarks to the Author):

The authors have replied in good faith to the criticisms from the referees.

Again, we thank the reviewers for their important comments and suggestions for improving the manuscript.